# Lung Fibrosis Is Improved by Extracellular Vesicles from IFNγ-Primed Mesenchymal Stromal Cells in Murine Systemic Sclerosis

**DOI:** 10.3390/cells10102727

**Published:** 2021-10-13

**Authors:** Pauline Rozier, Marie Maumus, Alexandre Thibault Jacques Maria, Karine Toupet, Christian Jorgensen, Philippe Guilpain, Danièle Noël

**Affiliations:** 1IRMB, University of Montpellier, INSERM, 34295 Montpellier, France; pauline.rozier@inserm.fr (P.R.); marie.maumus@inserm.fr (M.M.); a-maria@chu-montpellier.fr (A.T.J.M.); karine.toupet@inserm.fr (K.T.); christian.jorgensen@inserm.fr (C.J.); p-guilpain@chu-montpellier.fr (P.G.); 2Department of Internal Medicine, Multi-Organic Diseases, CHU, 34295 Montpellier, France; 3Clinical Immunology and Osteoarticular Disease Therapeutic Unit, Department of Rheumatology, CHU, 34295 Montpellier, France

**Keywords:** mesenchymal stromal cell, scleroderma, extracellular vesicles, exosomes, microvesicles, therapy

## Abstract

Background: Systemic sclerosis (SSc) is a severe autoimmune disease for which mesenchymal stromal cells (MSCs)-based therapy was reported to reduce SSc-related symptoms in pre-clinical studies. Recently, extracellular vesicles released by MSCs (MSC-EVs) were shown to mediate most of their therapeutic effect. Here, we aimed at improving their efficacy by increasing the MSC-EV dose or by IFNγ-priming of MSCs. Methods: small size (ssEVs) and large size EVs (lsEVs) were recovered from murine MSCs that were pre-activated using 1 or 20 ng/mL of IFNγ. In the HOCl-induced model of SSc, mice were treated with EVs at day 21 and sacrificed at day 42. Lung and skin samples were collected for histological and molecular analyses. Results: increasing the dose of MSC-EVs did not add benefit to the dose previously reported to be efficient in SSc. By contrast, IFNγ pre-activation improved MSC-EVs-based treatment, essentially in the lungs. Low doses of IFNγ decreased the expression of fibrotic markers, while high doses improved remodeling and anti-inflammatory markers. IFNγ pre-activation upregulated *iNos*, *IL1ra* and *Il6* in MSCs and ssEVs and the PGE2 protein in lsEVs. Conclusion: IFNγ-pre-activation improved the therapeutic effect of MSC-EVs preferentially in the lungs of SSc mice by modulating anti-inflammatory and anti-fibrotic markers.

## 1. Introduction

Systemic sclerosis (SSc) is an autoimmune disease with a severe prognosis due to generalized fibrosis and vasculopathy [1]. Currently, symptomatic management is the only strategy available to relieve patients and no curative treatment can reverse the disease. For some patients, immunosuppressive drugs and hematopoietic stem cell transplantation could be effective in stopping the course of the disease, but these also come with severe side effects [2]. More recently, mesenchymal stromal cells (MSCs) have demonstrated therapeutic benefit in preclinical models of SSc thanks to their pleiotropic properties. Currently, MSCs are being evaluated in the clinics, and promising results have been reported [3].

MSCs exert their therapeutic function through the release of many soluble mediators that are secreted in the extracellular milieu and/or within extracellular vesicles (EVs). EVs are a heterogeneous population of vesicles that are characterized by their size and biogenesis [4]. Apart from apoptotic bodies that are released by apoptotic cells, the two main EV subtypes are exosomes, which are below 150 nm in diameter and generated via the endosomal pathway inside multivesicular bodies, and microvesicles, or microparticles, whose size is above 120 nm and that form by budding of the plasma membrane. Because of overlapping sizes, current isolation procedures allow for the isolation of small size EVs (ssEVs) and large size EVs (lsEVs), which are enriched in exosomes and microvesicles, respectively [5]. Both types of EVs contain a cargo of proteins, lipids, and nucleic acids (including DNA, mRNA, miRNA) that mediate the functions of parental cells.

Using the HOCl-induced murine model of SSc, we previously reported that murine and human MSCs from bone marrow (BM) and adipose tissue (AT) are efficient to stop the course of the disease and prevent skin and lung sclerosis [6,7]. Recently, we demonstrated that EVs isolated from murine BM-MSCs and human AT-MSCs reproduce the therapeutic effect of parental cells [8]. We showed that both ssEVs and lsEVs powerfully stop disease progression and regulate expression of fibrotic and remodeling markers in HOCL-induced murine SSc. They also down-regulate inflammation in the skin and lungs of treated mice. The beneficial effect has been associated with the presence of miR-29a-3p in murine ssEVs and lsEVs as well as in total EVs from human AT-MSCs. MiR-29a-3p was shown to act via the regulation of type I and III collagens, apoptotic factors (*Bax*, *Bcl2*, *Bcl-xl*), methylation-regulating genes (*Tet1*, *Dnmt3a*) and *Pdgfrbb* in the skin of SSc mice. 

In the present study, we investigated the possibility of enhancing the therapeutic effect of ssEVs and lsEVs isolated from murine BM-MSCs (mMSC) in the murine model of SSc. We evaluated the interest of using a higher dose of EVs or the pre-activation of MSCs by IFNγ, which is known to stimulate their immunosuppressive effect, before EV isolation [9,10,11]. We also tested whether IFNγ pre-treatment of MSCs could up-regulate the production of immunosuppressive factors and their release within EVs, thereby enhancing their therapeutic effect in SSc. 

## 2. Materials and Methods

### 2.1. Mesenchymal Stromal Cell Expansion 

C57BL/6 mice-derived mMSCs were cultured in proliferative medium containing DMEM, 10% fetal calf serum (FCS), 100 μg/mL penicillin/streptomycin and 2 mmol/mL glutamine. Their characterization by phenotyping and tri-lineage differentiation potential has been reported before [12]. We used mMSCs between passage 12 and 18 for the following experiments.

### 2.2. Production and Isolation of EVs

EV subtypes were produced from mMSCs and characterized according to ISEV recommendations as previously described [8,13]. Briefly, mMSCs were seeded at 7000 cells/cm^2^ in proliferative medium for 24 h. After a wash with phosphate buffer saline (PBS), the production medium (DMEM containing 3% EV-free FCS) was added for 48 h. When necessary, recombinant mouse IFNγ (1 or 20 ng/mL) was added (R&D Systems, Bio-Techne, Noyal Chatillon, France). After removing cells and debris thanks to low speed centrifugation, lsEVs were pelleted by a first ultracentrifugation at 18 000× g, 4 °C, for 1 h and ssEVs were pelleted by a second ultracentrifugation at 100,000× *g*, 4 °C, for 2 h. Both pellets were then washed in PBS and submitted to another ultracentrifugation round at 100,000× *g* for 2 h. EV subtypes were characterized according to their morphology and their size thanks to cryoTEM and nanotracking analysis. Protein content was evaluated thanks to cytometry and western blot experiments, as reported elsewhere [8]. 

### 2.3. Animal Model and Histopathological Analysis

Mice were raised in the conditions required by the European guidelines for the care and use of laboratory animals (2010/63/UE). This project was approved by the Regional Ethics Committee on Animal Experimentation (APAFIS#53512016050919079187). SSc was induced onto six-week-old female BALB/cJRj mice (Janvier Labs, Le Genest-Saint-Isle, France) using daily HOCl intradermic injections for 42 days (150 µL into two sites at the base of the tail). Once a week, skin thickness was measured thanks to a caliper. At day 21, groups of 7–8 mice were formed to homogenize the mean skin thickness between groups. Mice received one intravenous injection (100 µL) of either NaCl 0.9% (control group), 250,000 mMSCs or 250 ng EVs (or 1500 ng depending on the experiment). Mice were split in the cages to minimize bias between cages. At day 42, blood, lungs and skin biopsies (6 mm punchs) were recovered. Blood was allowed to coagulate at room temperature (RT) at least 30 min and was then subjected to centrifugation at 1000× *g*, RT for 10 min to recover the serum, which was stored at −80 °C. Skin or lung samples for molecular analysis were stored at −80 °C. Samples dedicated to histology were directly fixed in 4% formaldehyde before paraffin embedding and routine histology. Histological slides (5 μm thick) were stained by Masson’s trichrome staining and analyzed with Nanozoomer (Hamamatsu) and NDP.view2 software to measure dermal thickness. 

### 2.4. RNA Extraction and RT-qPCR 

Skin and lung samples were grinded in RLT-buffer containing 1% β-mercaptoethanol and processed using the RNeasy Mini Kit for total RNA extraction, according to manufacturer’s recommendations (Qiagen, Courtaboeuf, France). RNA (500 ng) was reverse transcribed by M-MLV reverse transcriptase (ThermoFisher Scientific, Illkirch-Graffenstaden, France). Using specific primers, main fibrotic (Col1a1, Col3a1, Tgfβ1, TgfβR2), remodeling *(MMP9*, *MMP1/TIMP1)*, inflammatory *(IL1β*, *TNFα)* and oxidative stress (*Cox2*, *Hmox1*, *Sod2*) markers were analysed (Table 1). Real time PCR was performed on 20 ng cDNA using the Master SYBR Green I mix (Roche Diagnostics, Meylan, France). TATA binding protein (*Tbp*) was used as the housekeeping gene and data were expressed as *Tbp* relative expression and as fold change compared to control non-treated group using the formulae 2^−ΔCt^ or 2^−ΔΔCt^, respectively. 

MSC and EV samples were treated with Qiazol reagent (ThermoFisher Scientific) using miRNeasy Mini or Micro Kit, respectively (Qiagen, Courtaboeuf). Reverse transcription was performed using M-MLV Reverse Transcriptase kit (ThermoFisher Scientific), followed by real-time PCR on 10 ng cDNA using specific primers (Table 1) and SYBR Green I Master mix (Roche Diagnostics). Data were normalized to the expression of D-glyceraldehyde-3-phosphate dehydrogenase (*Gapdh*) housekeeping gene for mMSCs and expressed using the formulae 2^−ΔCT^. For miRNA, reverse transcription of 10 ng total RNA was performed according to the recommendations of TaqMan^®^ MicroRNA Reverse Transcription Kit (ThermoFisher Scientific). Real-time PCR was performed on 2 ng cDNA using Taqman microRNA hsa-miR29a-3p (Life Technologies, Villebon-sur-Yvette, France) and Taqman Master Mix II no UNG (Life Technologies). Data were normalized to the expression of U6 snRNA (Life Technologies) for mMSCs and data were expressed as 100/Ct for EVs.

#### 2.4.1. Measure of Advanced Oxidation Protein Products

Sera or a range of 200 µL of standard chloramine T solution (0 to 1000 μM) were incubated with 20 μL acetic acid and 10 µL potassium iodide (1.16 M) at RT. Optical densities were measured at 340 nm on a microplate reader (Varioskan, ThermoFisher Scientific), before incubation and each minute for 10 min. AOPP concentrations were expressed as chloramine T equivalents (μM).

#### 2.4.2. Measure of Anti-Oxidant Capacity

Anti-oxidant capacity (AOC) was measured on sera diluted to 1/10 or a standard range of Trolox using the Antioxidant Assay Kit (Cayman Chemical, Interchim, Montluçon, France). The absorbance was read at 750 nm and AOC was expressed as mM Trolox equivalents.

### 2.5. ELISA

After the last step of centrifugation, EVs were suspended in 100 µL of NP40 cell lysis buffer (Invitrogen, ThermoFisher Scientific). Total proteins were quantified using MicroBCA assay kit (ThermoFisher Scientific). Samples were stored at −20 °C until quantification using DuoSet^®^ ELISA kits for HGF, IL1RA, IL6, TGFβ1 (R&D Systems) and prostaglandin E2 (PGE2) using Multi-format ELISA kit (Arbor Assays, Clinisciences, Nanterre) on 1 µg of total proteins.

### 2.6. Analysis of miRNA Profiles

The miRNA profile of EVs was analysed by HTG Next-Generation Sequencing using an Illumina NextSeq 500 system by the company Firalis (Huningue, France). We selected the 277 miRNAs that were common to human and mouse species and normalized the cpm value of each miRNA on the total cpm count. We then selected the 131 miRNAs that had a cpm value > 500.

Expression of miRNAs in IFNγ-pre-activated ssEVs and lsEVs were normalized to their expression in non-activated ssEVs and lsEVs. We selected the miRNAs that were up- and down-regulated (FC > 1.5 and FC < −1.5, respectively) in IFNγ-pre-activated ssEVs and lsEVs.

### 2.7. Gene Ontology Pathway Analysis

The identification of validated target genes modulated by miRNAs in naive or IFNγ-pretreated lsEVs and ssEVs were done using TaRbase [14]. Enrichr and Panther software were used to perform biological pathway enrichment analysis and Gene Ontology (GO) enrichment analysis for identification of gene enrichment in cellular component and molecular function [15,16,17].

### 2.8. Statistical Analysis

Statistical analyses were performed using GraphPad 8 Prism Software. Data distribution was evaluated using the Shapiro-Wilk normality test. Each single group was compared to the control group using the Student t-test or the Mann-Whitney test when values were parametric or non-parametric, respectively. For values normalized to 1, a one sample t-test or Wilcoxon test were performed when values assumed or not a normal distribution, respectively. Data are presented as mean ± SEM.

## 3. Results

### 3.1. A High Dose of MSC-EVs Was Not Beneficial to SSc Mice

Isolation and characterization of ssEVs and lsEVs from murine BM-MSCs have been described previously [8]. After systemic injection of these MSC-derived ssEVs and lsEVs, we showed that the two sub-populations of EVs can reduce clinical symptoms and histopathological alterations in SSc. With the aim to improve treatment efficacy, we tested whether a higher dose of lsEVs might be beneficial. By comparison with the dose of 250 ng used in the previous study, a high dose of lsEVs (1500 ng) did not stop disease progression as shown by a continuous skin thickness increase, which was similar to non-treated mice, and a higher skin thickness measured at day 42 (Figure 1A,B). 

On histological sections, the dermal thickness in mice receiving the high dose of lsEVs was similar to non-treated mice and higher than in mice receiving the low dose of lsEVs (Figure 1C,D). Contrary to the low dose, the high dose of lsEVs did not significantly improve the expression of several fibrotic (*Col1a1*), remodeling (*Mmp9*) and inflammatory (*Il1β*, *Tnfα*) markers (Figure 1E). Expression of *αSma* and *Il1β* was higher in mice treated with the high dose versus the low dose of lsEVs while the anti-oxidant marker *Sod2* was lower. Similar results were observed in the lungs. On histological sections, no improvement in pulmonary fibrosis was noticed as indicated by a dense and infiltrated parenchyma in mice injected with the high dose of lsEVs (Appendix A). Expression of fibrotic markers (*Col1a1*, *Col3a1*, *αSma*), did not improve although the expression of *Mmp9* and *Il1β* was reduced to similar levels as the low dose of lsEVs (Appendix A). Of note, the mRNA levels of *Mmp1* (as illustrated by the ratio *Mmp1/Timp1*), *Hmox1*, *Nfe2l2* and *Sod2* increased after the injection of the high dose of lsEVs. Altogether, the results indicated that the high dose of lsEVs did not improve the clinical and histological features of SSc, even though remodeling and anti-oxidative capacity improved in the lungs.

### 3.2. Low Dose IFNγ Pre-Activation Improved the Anti-Fibrotic Effect of MSC-EVs in Lungs 

We then evaluated whether the pre-activation of MSCs by a low dose of IFNγ (1 ng/mL) could improve the therapeutic effect of ssEVs and lsEVs, notably their anti-inflammatory effect [18]. Although disease progression was stopped or slowed down in all treated groups, pre-activated EV subtypes (ssEV A1 and lsEV A1) were less efficient than non-activated EV subtypes (ssEV NA and lsEV NA) (Figure 2A). At day 42, the skin thickness of mice that received ssEV A1 and lsEV A1 was significantly thicker as compared to non-activated EVs (Figure 2B). The measure of dermal thickness on skin histological sections indicated a lower thickness in all groups and revealed a heterogeneity of response in the ssEV A1 group (Figure 2C,D). All the fibrotic and inflammatory markers were significantly lower in the treated groups, even though lsEV A1 were less efficient than lsEV NA to reduce the expression of *Tgfβ1*, *αSma* and *Tnfα* (Figure 2E). As expected, the gelatinase *Mmp9* was reduced in all treated groups and the *Mmp1/Timp1* ratio, which is representative of matrix remodeling, was increased. We also measured the advanced oxidized protein products (AOPP) concentration in the sera as a marker of oxidative stress. Interestingly, AOPP levels were significantly lower in all treated groups (Figure 2F).

Again, lsEV A1 were less efficient than lsEV NA to decrease AOPP levels. The anti-oxidative capacity (AOC) in the sera of treated groups was not increased as compared to that of the control group and was even decreased in the ssEV A1 group (Figure 2F). As a result, the AOPP/AOC ratio was significantly reduced in all treated groups, indicating an anti-oxidative effect of EVs, but to a lesser extend in the lsEV A1 group. 

In the lungs, the analysis of histological sections showed fewer collagen deposits in treated groups, regardless of EV subtype (Figure 3A). 

At the molecular level, both ssEV A1 and lsEV A1 decreased the expression of fibrotic markers (*Col3a1*, *Tgfβ1*, *Tgfβr2)* as compared to ssEV NA and lsEV NA or control group (Figure 3B). Nevertheless, ssEV A1 and lsEV A1 did not reduce the expression of *Mmp9* and the *Mmp1/Timp1* ratio was unchanged. The inflammatory markers *Il1β* and *Tnfα* were lower in all treated groups. Overall, MSC pre-activation did not improve the efficacy of the two EV subtypes on cutaneous fibrosis but improved at least partially lung fibrosis by reducing the expression of all the fibrotic and inflammatory markers.

### 3.3. High Dose IFNγ Pre-Activation Improved Remodeling and Anti-Inflammatory Effect of MSC-EVs in Lungs 

With the hope to increase these anti-fibrotic and anti-inflammatory effects of MSC-EVs in the lungs and possibly in the skin, we tested a higher dose of IFN-γ (20 ng/mL) to pre-activate MSCs. Both pre-activated MSC-EV subtypes (ssEV A20 and lsEV A20) stopped the disease course and reduced the skin thickness to similar levels as non-activated EV subtypes, but lsEV A20 were more efficient than ssEV A20 (Figure 4A,B). Histological sections of skin revealed a lower dermal thickness in all treated groups (Figure 4C). Most of the molecular markers improved in the skin of treated groups except in the ssEV A20 group for *Col1α1*, *Tgfβr2*, *Mmp1/Timp1*, *Il1β*, *Tnfα* (Figure 4D). Altogether, independently on pre-activation, lsEVs seemed more efficient than ssEVs, in particular on remodeling and inflammatory markers in the skin. 

In the lungs, improvement of several fibrotic, remodeling and inflammatory markers was observed in all treated groups even though statistical significance was not reached for some markers in this specific experiment (Figure 4E). The pre-activated ssEVs A20 and lsEVs A20 seemed to be more potent than non-activated EVs to improve *αSma*, *Tgfβr2* as well as all remodeling and inflammatory mediators (Figure 4E). No further beneficial effect was observed with lsEV A20 as compared to ssEV A20. Overall, EVs isolated from MSCs pre-activated with a high dose of IFNγ tended to improve all molecular markers in the lungs. 

### 3.4. IFNγ Pre-Activation Up-Regulated Anti-Inflammatory Factors in MSCs and MSC-EVs 

To better understand the mechanism of action of MSC-EVs, we compared the miRNA profile of EVs isolated from non-activated MSCs and MSCs pre-treated with a high dose of IFNγ (Table 2). 

We selected the miRNAs that were up- and down-regulated by a fold change > 1.5 and found that most of the modulated miRNAs were down-regulated by IFNγ (Figure 5A). Surprisingly, no miRNA was down-regulated in both ssEVs and lsEVs. We then performed a functional enrichment analysis on experimentally validated target genes. Using miRTarBase, we identified 27,979 target genes that were analyzed for biological pathway enrichment using Enrichr software. The first overrepresented biological pathway was related to IFNγ (Figure 5B). Target genes were also classified into distinct functional categories, including molecular functions and cellular components, using GO Term Enrichment Analysis (Figure 5C). 

We then quantified the expression of known immunosuppressive and anti-fibrotic markers in MSCs. At least one of the two doses of IFNγ up-regulated *Il1ra*, *Il6 and Cox2* mRNAs in MSCs while the expression of *iNos*, *Tsg6*, *Hgf* and *Tgfβ1* was not significantly modulated although *iNos* and *Tsg6* mRNA increased by a 20-fold and 2-fold factor, respectively (Figure 6A). 

Expression of miR-29a-3p, which was shown to be involved in the beneficial effect of MSC-derived EVs in SSc [8], tended to be up-regulated in MSCs pre-activated by the high dose of IFN*γ*. Those markers were then quantified at the mRNA level in ssEV A20 and lsEV A20. No difference in the mRNA levels of the different markers was observed between ssEVs and lsEVs from non-activated and high dose IFNγ-pre-activated MSCs (Figure 6B). Nevertheless, a tendency to higher levels of *iNos*, *Il1ra* and *Il6* were observed in ssEV A20 compared to ssEV NA, suggesting that part of the mRNA from the parental cells were packaged in ssEVs. At the protein level, IL6 and TGFβ1 were not detected in any types of EVs while IL1-RA and PGE2 were slightly higher in lsEVs than ssEVs (Figure 6C). By contrast, HGF levels were lower in both non-activated or IFN*γ* pre-activated lsEVs compared to ssEVs. In summary, IFN*γ* pre-activation up-regulated the expression of several anti-inflammatory factors in mMSCs that were not found in EVs, suggesting that other factors might be related to the improvement of SSc markers in the lungs.

## 4. Discussion

In the present study, we report that IFNγ pre-activation of MSCs improved the therapeutic effect of EVs in the lungs of SSc-induced mice with no major impact on the skin. 

We first showed that the therapeutic efficacy of MSC-EVs was dose-dependent. Interestingly, increasing the dose of lsEVs by a six-fold factor abolished the beneficial effect of EVs. This is consistent with our previous study that demonstrated that increasing the quantity of systemically injected MSCs from 2.5 × 10^5^ cells to 1 × 10^6^ cells reversed the beneficial role of cell therapy [6]. The reason for this is not known. This might be related to the local accumulation of EVs that might contribute to a dysregulated crosstalk with surrounding endogenous cells in a context of oxidative stress and inflammation related to SSc. This might also be related to the raise of a number of factors in serum, including inflammatory cytokines or profibrotic factors to levels that can induce unwanted effects on organs or immune cells and counteract the beneficial effects. The only markers that improved following infusion of the high lsEV dose were anti-oxidative stress genes in lungs. Whether this reflects an adaptive response to the elevation of oxidative stress in the lungs of mice receiving the high dose of lsEVs remains to be elucidated. Nevertheless, the data indicate that the optimal dose will have to be determined in larger preclinical models before clinical translation.

One important finding of the study is the interest in using EVs from IFNγ pre-activated MSCs to enhance their beneficial effect in the lungs of SSc mice. It has already been shown that MSC priming by hypoxia or genetic modification may enhance their therapeutic properties in other applications [19,20]. Improvement of fibrotic, remodeling and inflammatory markers was observed by comparison with EVs from non-activated MSCs. Few studies have investigated the interest of IFNγ pre-activation of MSCs before EV isolation. Nevertheless, EVs isolated from IFNγ pre-activated human umbilical cord-derived MSCs were shown to increase the survival of rats with *Escherichia coli*-induced pneumonia and to reduce the lung injury [21]. EVs from IFNγ pre-activated human BM-MSCs reduced experimental autoimmune encephalomyelitis (EAE) via the generation of Treg cells [22]. By contrast, EVs from primed human cord blood-derived MSCs were not able to protect kidney from ischemia-reperfusion injury or displayed similar efficacy as naïve EVs in experimental spinal cord injury [23,24]. In vitro, EVs isolated from human MSCs, whether pre-activated or not with IFNγ, exerted similar ability to inhibit T-cell proliferation although only pre-activated EVs contained IDO mRNA [25] and a number of anti-inflammatory RNAs and proteins [22]. In addition to IDO and iNOS in human and mouse MSCs, respectively, up-regulation of PDL1 or CD200 was also proposed to sustain the enhanced immunosuppressive function of IFNγ pre-activated MSC-EVs [9]. In our study, as expected, we found an up-regulation of several known immunosuppressive markers in IFNγ pre-activated MSCs. The mRNAs of several of those markers tended to be up-regulated in ssEV A20 and might be related to the improvement of SSc-associated markers in the lungs. These markers were not up-regulated in lsEV A20, suggesting that they are differentially routed according to the biogenesis pathways of EVs. Furthermore, these results suggest that EVs can probably transfer mRNAs to target tissues where they can be translated into effector proteins regulating intrinsic pathways in the recipient cells. Interestingly, the mRNA levels of *Cox2* were down-regulated in both ssEV A20 and lsEV A20 whereas the quantity of the protein PGE2 was not affected. Nevertheless, the quantity of PGE2 was higher in lsEVs compared to ssEVs, while HGF was lower. We did not found a higher content of miR-29a-3p in lsEVs, which is consistent with the limited effect of IFNγ pre-activation on the miRNA landscape of human MSC-EVs [26]. The differential effect of EVs in the skin and lungs of SSc mice might be related to the route of administration and a higher effect in the lungs, which are primarily targeted using systemic administration. Another hypothesis is the up-regulation of specific soluble and membrane markers, which could impact their biodistribution and act differentially on different organs. To our knowledge, no differential effect of pre-activated versus naïve MSC-EVs has been described on organ functions so far and therefore a better understanding of IFNγ-pre-activation on EV functions is needed.

Beside functional differences between EVs isolated from naïve and pre-activated MSCs, we also observed a slightly but constant higher therapeutic effect of lsEV A20 compared to ssEV A20 in the skin and lungs of treated mice. Controversial data are reported from the literature. SsEVs have been previously shown to outperform lsEVs in collagen-induced arthritis, kidney injury or in delayed-T hypersensitivity mouse models [27,28,29]. In another model, the proportion of ssEVs versus lsEVs was decreased after IFNγ pre-activation and related to a concomitant loss of therapeutic efficiency, suggesting that lsEVs were more efficient than ssEVs [23]. Differences in protein, mRNA and miRNA content were proposed to explain the differential regenerative capacities of EV subtypes in a model of acute kidney injury [27]. The proproliferative effect of ssEVs was ascribed to the presence of many factors playing a role in the maintenance of cell cycle, while p53, a negative regulator of cell cycle, was found in lsEVs. Differences in membrane markers expressed by EV subtypes could also explain differential targeting of injured or diseased tissue since MSC-EVs rapidly localized in the injured organs and remained up to seven days after systemic administration [30]. The different cargo of ssEVs and lsEVs, and their inherent functions can differently impact the targeted tissues according to the disease. Further studies are therefore needed to decipher the respective roles of ssEVs and lsEVs, keeping in mind that the current isolation protocol does not allow for the isolation of pure populations of exosomes or microvesicles. 

In conclusion, we showed that IFNγ pre-activation of MSCs enhanced the beneficial effect of ssEVs and lsEVs by regulating several markers whose expression is altered in SSc. We further observed that IFNγ pre-activated lsEVs might be more efficient than ssEVs in this specific disease application.

## Figures and Tables

**Figure 1 cells-10-02727-f001:**
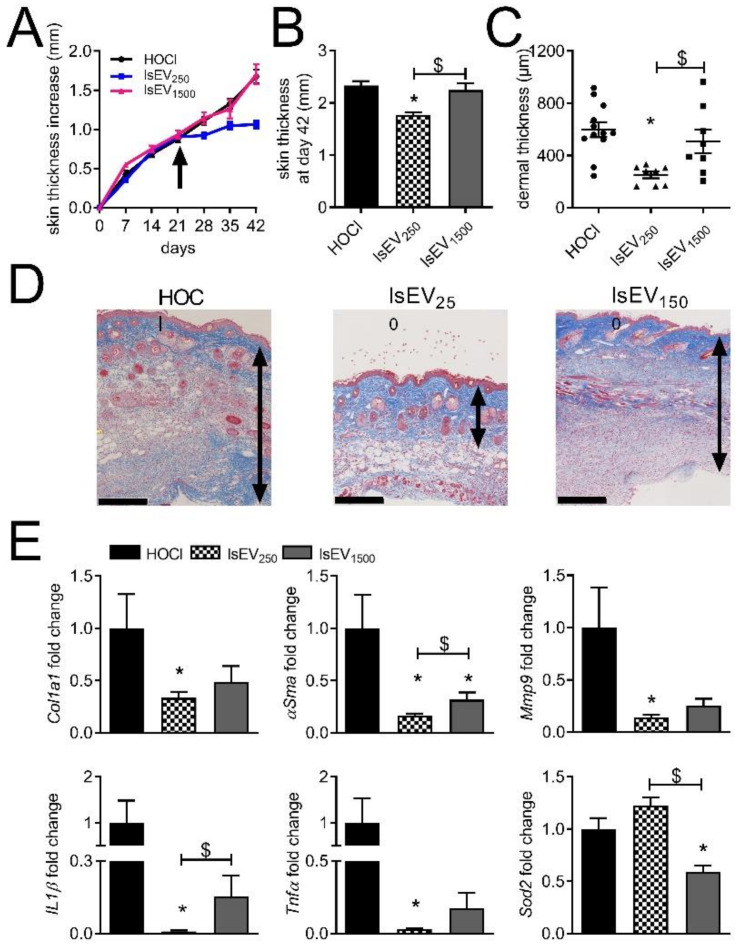
Dose effect of lsEVs isolated from MSCs in the murine model of HOCl-induced SSc. (**A**) Measures of the skin thickness increase in control mice (HOCl) and mice that have been injected with 250 or 1500 ng of large size extracellular vesicles (lsEV_250_ or lsEV_1500_, respectively) on day 21 (arrow). (**B**) Mean skin thickness in the different groups of mice at day 42. (**C**) Mean dermal thickness on histological sections of skin from the three groups of mice. (**D**) Photographs of representative histological sections of skin after Masson’s trichrome staining (the double arrow delineates the dermis; scale bar, 250 µm). (**E**) Gene expression in skin samples as expressed as fold change in treated versus HOCl control mice. Data are presented as mean ± SEM (*n* = 8 to 12 per group; *: *p* < 0.05 versus control or $: *p* < 0.05 versus the indicated group of mice).

**Figure 2 cells-10-02727-f002:**
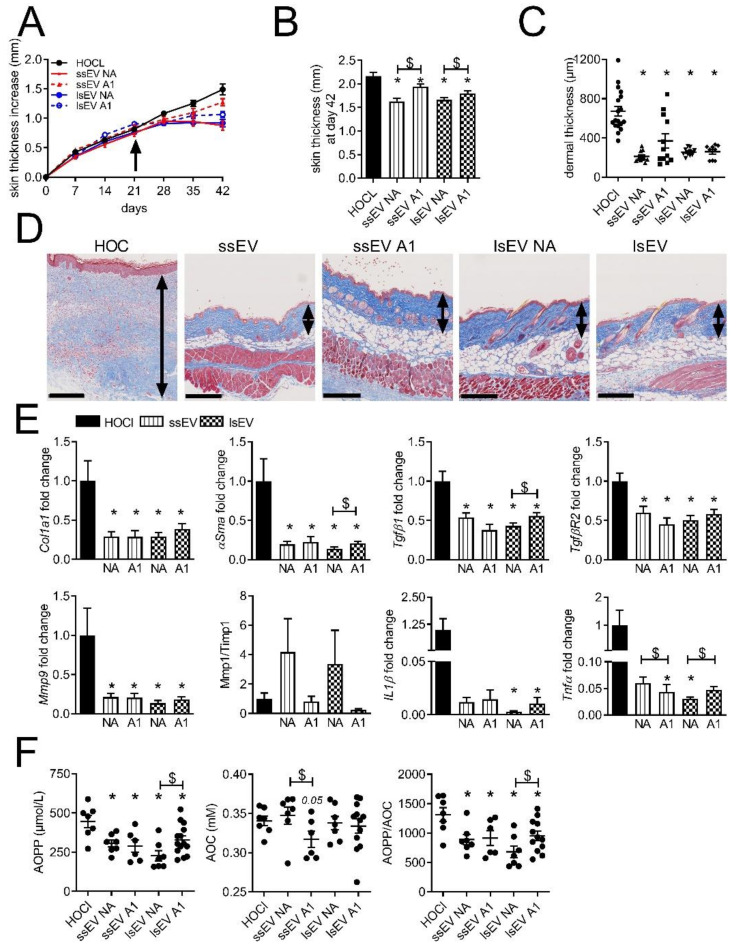
Therapeutic effect of EVs isolated from MSCs pre-activated with low dose of IFNγ in skin from HOCl-induced SSc. (**A**) Measures of the skin thickness increase in control mice (HOCl) and mice that have been injected with small size or large size extracellular vesicles isolated from non-activated MSCs (ssEV NA or lsEV NA) or MSCs pre-activated by 1 ng/mL IFNγ (ssEV A1 or lsEV A1) on day 21 (arrow). (**B**) Mean skin thickness in the different groups of mice at day 42. (**C**) Mean dermal thickness on histological sections of skin from the different groups of mice. (**D**) Photographs of representative histological sections of skin after Masson’s trichrome staining (the double arrow delineates the dermis; scale bar, 250 µm). (**E**) Gene expression in skin samples as expressed as fold change in treated versus HOCl control mice. (**F**) Quantification of advanced oxidation protein products (AOPP) and of anti-oxidant capacity (AOC) in the serum of mice as expressed as chloramine T and Trolox equivalents, respectively, and expression of the AOPP/AOC ratio. Data are presented as mean ± SEM (*n* = 12 to 18 per group from 2 separate experiments (except for data in f); *: *p* < 0.05 versus control or $: *p* < 0.05 versus the indicated group of mice).

**Figure 3 cells-10-02727-f003:**
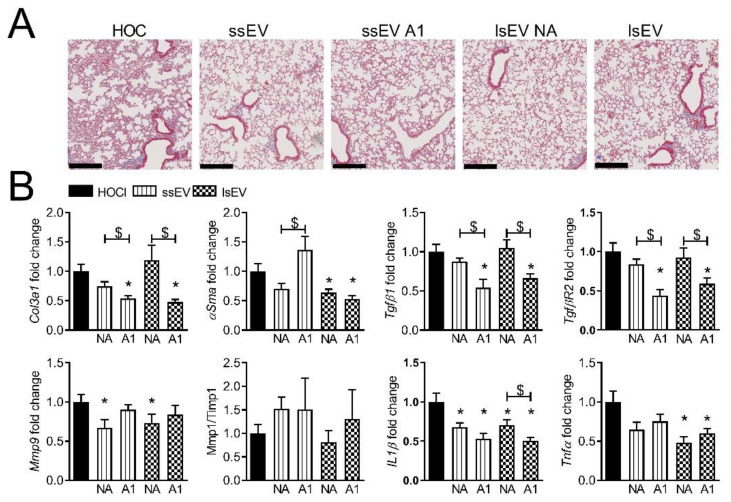
Therapeutic effect of EVs isolated from MSCs pre-activated with low dose of IFNγ in lungs from HOCl-induced SSc. (**A**) Photographs of representative histological sections of lungs after Masson’s trichrome staining in control mice (HOCl) and mice that have been injected with small size or large size extracellular vesicles isolated from non-activated MSCs (ssEV NA or lsEV NA) or MSCs pre-activated by 1 ng/mL IFNγ (ssEV A1 or lsEV A1) (scale bar, 250 µm). (**B**) Gene expression in lung samples as expressed as fold change in treated versus HOCl control mice. Data are presented as mean ± SEM (*n* = 12 to 18 per group from two separate experiments (except for data in f); *: *p* < 0.05 versus control or $: *p* < 0.05 versus the indicated group of mice).

**Figure 4 cells-10-02727-f004:**
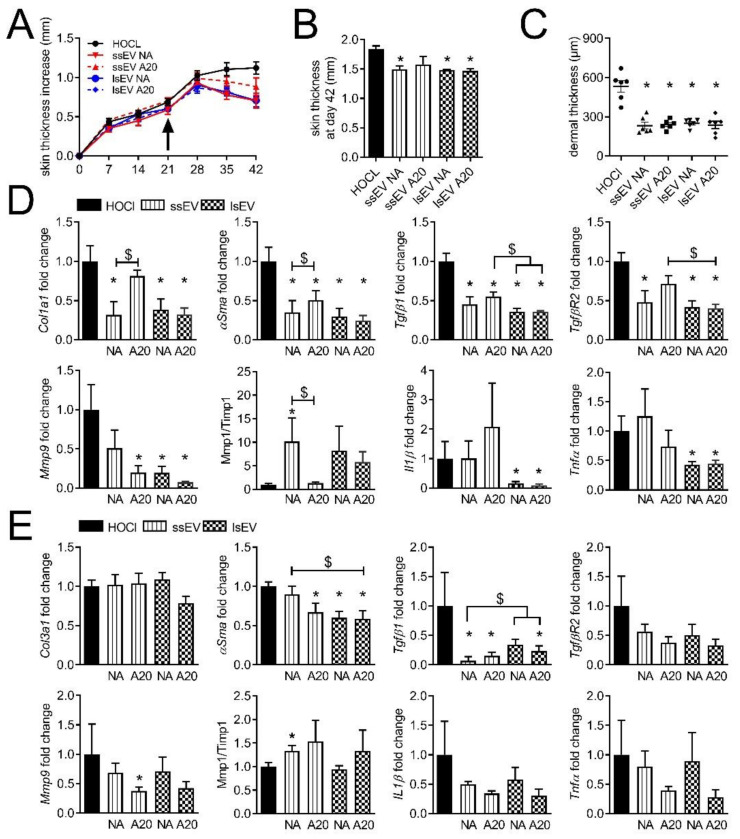
Therapeutic effect of EVs isolated from MSCs pre-activated with high doses of IFNγ in the murine model of HOCl-induced SSc. (**A**) Measures of the skin thickness increase in control mice (HOCl) and mice that have been injected with small size or large size extracellular vesicles isolated from non-activated MSCs (ssEV or lsEV NA) or MSCs pre-activated by 20 ng/mL IFNγ (ssEVs or lsEVs A20) on day 21 (arrow). (**B**) Mean skin thickness in the different groups of mice at day 42. (**C**) Mean dermal thickness on histological sections of skin from the different groups of mice. (**D**) Gene expression in skin samples as expressed as fold change in treated versus HOCl control mice. (**E**) Gene expression in lung samples as expressed as fold change in treated versus HOCl control mice. Data are presented as mean ± SEM (*n* = 6 per group; *: *p* < 0.05 versus control or $: *p* < 0.05 versus the indicated group of mice).

**Figure 5 cells-10-02727-f005:**
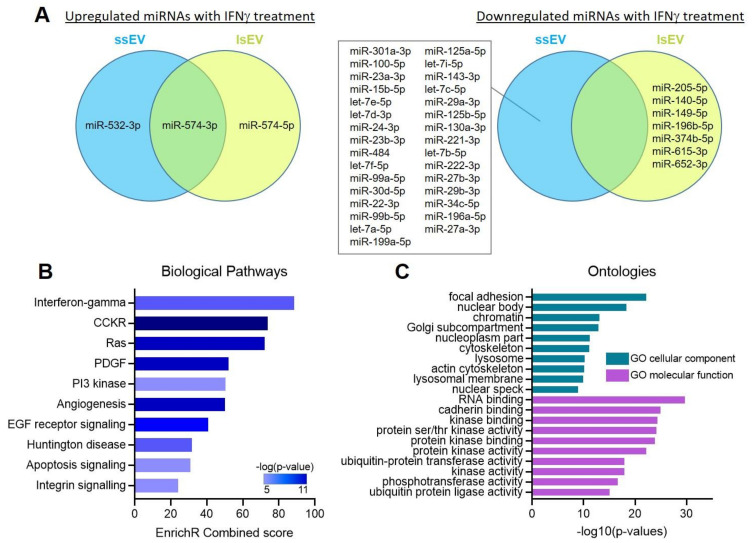
miRNAs modulated in MSC-EV subtypes after pre-activation with IFNγ. (**A**) Venn diagrams of the miRNAs that are up- and down-regulated in ssEVs A20 or lsEVs A20 (left and right panels, respectively) compared to ssEV NA or lsEV NA. (**B**) Gene Ontology enrichment analysis of biological pathways for the target genes of miRNAs modulated with IFNγ from (**A**) with a fold change>1.5. (**C**) Gene Ontology enrichment analysis of biological pathways for cellular components and molecular functions of the target genes of miRNAs modulated with IFNγ from (**A**).

**Figure 6 cells-10-02727-f006:**
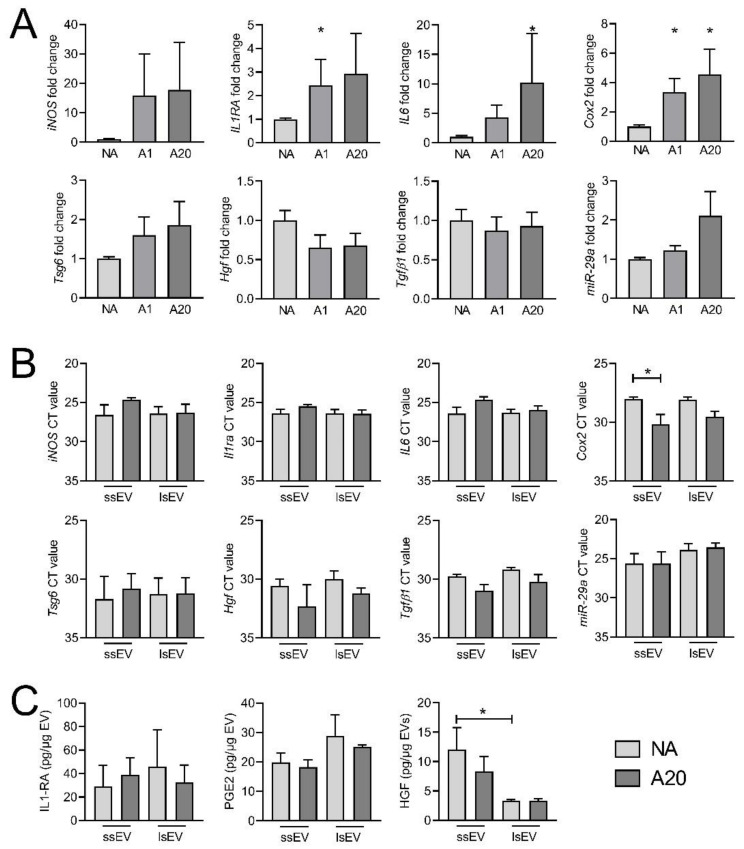
Immunosuppressive factors up-regulated in MSCs and in MSC-EVs after pre-activation with IFNγ. (**A**) Gene expression of known immunosuppressive and anti-fibrotic markers in non-activated MSCs (NA) or MSCs pre-activated with 1 (A1) or 20 ng/mL (A20) IFNγ. (**B**) Gene expression of known immunosuppressive and anti-fibrotic markers in ssEV NA, lsEV NA, ssEV A20 and lsEV A20). (**C**) Protein quantification of known immunosuppressive and anti-fibrotic markers in ssEV NA, lsEV NA, ssEV A20 and lsEV A20 obtained by ELISA. Data are presented as mean ± SEM (*n* = 3); *: *p* < 0.05 versus NA mMSCs or the indicated group control.

**Table 1 cells-10-02727-t001:** List of primers for gene analysis by RT-qPCR.

Gene Name	Sequence Forward	Sequence Reverse
*Acta2(αSma)*	AAGGCCAACCGGGAGAAAAT	AGCCAAGTCCAGACGCATGA
*Col1a1*	TGTTCAGCTTTGTGGACCTC	TCAAGCATACCTCGGGTTTC
*Col3a1*	CGGTGAACGGGGCGAAGCTGGTT	GACCCCTTTCTCCTGCGGCTCCT
*Cox2*	GCATTCTTTGCCCAGCACTT	AGACCAGGCACCAGACCAAAGA
*Gapdh*	GGTGCTGAGTATGTCGTGGA	GTGGTTCACACCCATCACAA
*Hgf*	TGCCCTATTTCCCGTTGTGA	CGCTTCTCCTCGCCTCTCTC
*Hmox1*	GCAGAGCCGTCTCGAGCATA	GCATTCTCGGCTTGGATGTG
*Il1ra*	AGGCCCCACCACCAGCTTTGA	GGGGCTCTTCCGGTGTGTTGGT
*Il1β*	TTTGACAGTGATGAGAATGACCTGTTC	TCATCAGGACAGCCCAGGTCAAAG
*Il6*	TGGGACTGATGCTGGTGACA	TTCCACGATTTCCCAGAGAACA
*iNos*	CCTTGTTCAGCTACGCCTTC	GCTTGTCACCACCAGCAGTA
*Mmp1*	TTCAAAGGCAGCAAAGTATGGGCT	CCAGTCTCTTCTTCACAAACAGCAGCA
*Mmp9*	TCCAGTTTGGTGTCGCGGAGCACG	CAGGGGGAAAGGCGTGTGCCAGA
*Nfe2l2*	CGCCAGCTACTCCCAGGTTG	ACTTTCAGCGTGGCTGGGGA
*Sod2*	TCAGGACCCATTGCAAGGAA	TGTGGCCGTGAGTGACGTTT
*Tbp*	GGGAGAATCATGGACCAGAA	CCGTAAGGCATCATTGGACT
*Tgfβ1*	TGCGCTTGCAGAGATTAAAA	CTGCCGTACAACTCCAGTGA
*TgfβR2*	CGACCCCAAGCTCACCTACC	CAACAACAGGTCGGGACTGC
*Timp1*	CTCCGCCCTTCGCATGGACATT	GGGGGCCATCATGGTATCTGCTCT
*Tnfα*	AGCCCACGTCGTAGCAAACCA	TGTCTTTGAGATCCATGCCGTTGGC

**Table 2 cells-10-02727-t002:** List of modulated miRNAs.

	MSC		ssEV		lsEV	
	−IFNγ	+IFNγ	−IFNγ	+IFNγ	−IFNγ	+ IFNγ
let-7a-5p	39,871	33,617	10,128	5963	21,047	21,213
let-7b-5p	47,447	35,255	8648	4556	21,621	21,476
let-7c-3p	403	0	0	0	0	0
let-7c-5p	57,948	42,747	11,281	6154	26,131	26,347
let-7d-3p	1097	879	1585	1000	1544	1455
let-7d-5p	17,079	16,250	5002	3356	9475	8869
let-7e-3p	253	0	0	0	0	0
let-7e-5p	7938	6261	1621	1023	4224	4091
let-7f-1-3p	245	114	0	0	0	0
let-7f-5p	21,504	21,024	8136	4975	15,443	15,217
let-7g-5p	6287	7170	3143	2408	4563	4024
let-7i-5p	33,329	30,098	8184	4723	16,303	13,232
miR-100-5p	7165	7546	2619	1675	5070	4458
miR-101-3p	513	1572	1757	2085	1188	1221
miR-103a-2-5p	367	168	0	0	0	0
miR-103a-3p	2655	2334	1695	1429	1978	1613
miR-106b-3p	521	275	0	631	0	0
miR-106b-5p	1881	2134	3213	3287	2228	2239
miR-107	1667	1684	1127	1070	1170	0
miR-1247-5p	2752	1116	0	598	1544	1207
miR-1249	1213	626	4544	4571	3003	2429
miR-125a-3p	745	328	0	0	0	0
miR-125a-5p	13,783	15,755	5304	3068	11,867	10,808
miR-125b-1-3p	3316	944	0	0	0	0
miR-125b-5p	55,996	50,774	15,833	8527	31,962	31,144
miR-126-3p	0	0	4183	4440	1692	1555
miR-126-5p	0	0	2105	2089	1070	0
miR-128-3p	454	424	0	668	0	0
miR-130a-3p	7587	9334	3461	1860	5813	5799
miR-130b-3p	1339	2413	1193	821	2200	2280
miR-140-5p	1309	1419	0	0	1333	0
miR-142-5p	0	0	5341	5417	2163	1799
miR-143-3p	27,877	31,702	14,591	8415	25,698	21,680
miR-144-3p	0	0	12,449	18,149	3434	2808
miR-145-5p	37,004	33,556	16,311	8731	36,226	35,290
miR-146b-5p	4779	7806	1639	1519	3429	4867
miR-148a-3p	2180	2622	1247	903	1602	2148
miR-148b-3p	1009	974	0	728	0	0
miR-149-5p	1796	720	0	0	1292	0
miR-150-5p	0	0	3595	3486	1920	1511
miR-152-3p	1854	2393	0	0	1209	1746
miR-15a-5p	9268	11,252	8791	6577	10,976	7963
miR-15b-5p	10,225	11,817	7676	4862	12,178	8735
miR-16-5p	23,877	29,760	24,744	19,313	33,102	24,885
miR-17-5p	3092	3465	3345	2828	3773	3677
miR-181a-3p	270	0	0	0	0	0
miR-181a-5p	2155	1557	1241	1021	1466	1414
miR-181b-5p	944	634	0	0	0	0
miR-181d-5p	714	473	0	0	0	0
miR-183-5p	527	365	0	0	0	0
miR-185-5p	610	540	0	632	0	0
miR-186-5p	1164	1372	1963	1844	1696	1709
miR-18a-5p	437	457	0	0	0	0
miR-191-5p	3574	3677	3092	2965	3710	4040
miR-193a-3p	3563	2401	0	587	2061	2227
miR-195-5p	374	804	0	0	1055	0
miR-196a-5p	5952	4730	1297	602	3301	3194
miR-196b-3p	815	223	0	0	0	0
miR-196b-5p	2520	2101	0	0	1290	0
miR-199a-5p	15,996	17,854	6145	3584	13,123	13,706
miR-19a-3p	1182	1504	2363	2298	2326	1862
miR-19b-3p	6114	5510	6119	5431	6073	5334
miR-204-3p	992	285	0	578	0	0
miR-205-5p	0	117	0	0	1655	0
miR-20a-5p	1715	2219	2308	1860	2061	2192
miR-20b-5p	1504	1628	1924	1286	2043	1913
miR-210-3p	7921	5347	1569	1288	3041	3329
miR-214-5p	772	860	0	0	0	0
miR-21-5p	56,378	115,717	50,802	29,227	84,586	76,286
miR-218-5p	1364	3301	0	0	942	1318
miR-221-3p	60,280	36,507	17,469	9313	37,071	32,350
miR-222-3p	45,874	33,045	13,637	6941	31,186	26,064
miR-223-3p	0	0	10,435	11,464	3899	3621
miR-22-3p	30,464	35,897	9711	5829	19,711	16,146
miR-22-5p	831	857	0	0	0	0
miR-23a-3p	14,530	15,288	9451	6039	17,685	18,471
miR-23a-5p	1364	318	0	0	1541	1513
miR-23b-3p	9879	11,605	7103	4374	12,388	13,316
miR-24-3p	14,019	13,811	5365	3360	10,476	10,575
miR-25-3p	3410	3478	4018	4686	3312	3240
miR-26a-5p	14,943	24,472	11,473	8941	16,360	14,933
miR-26b-5p	4340	5251	5304	4160	5172	4732
miR-27a-3p	9509	9645	11,580	5186	18,358	13,643
miR-27a-5p	542	183	0	0	0	0
miR-27b-3p	3745	5331	5673	2863	10,455	7796
miR-28-5p	2498	2798	0	0	1707	1785
miR-29a-3p	32,286	36,765	14,087	7669	29,072	29,852
miR-29a-5p	442	545	0	0	0	0
miR-29b-3p	33,224	43,943	15,536	7714	27,775	26,096
miR-29c-3p	1299	3281	1722	1594	3460	3008
miR-301a-3p	1613	1329	1057	687	1350	1342
miR-30a-3p	431	298	0	0	0	0
miR-30a-5p	3415	4477	2124	1653	3523	3333
miR-30b-5p	3544	4877	4024	3000	5261	4684
miR-30c-2-3p	253	122	0	0	0	0
miR-30c-5p	5151	7006	4198	2929	5608	4956
miR-30d-5p	4961	5492	2658	1607	4626	4038
miR-30e-3p	464	299	0	0	0	0
miR-30e-5p	1188	1537	1222	953	1537	1351
miR-320a	2177	1001	0	576	1707	1609
miR-324-5p	970	620	0	0	0	0
miR-328-3p	1955	975	0	1194	1092	1342
miR-331-3p	829	642	0	0	0	0
miR-33a-5p	2930	2128	0	625	1409	1453
miR-342-3p	0	154	1680	1424	0	0
miR-34a-5p	2393	2012	0	0	1242	1193
miR-34c-5p	8039	7379	2217	1038	4598	3652
miR-361-5p	304	478	0	0	0	0
miR-374b-5p	685	749	0	0	1272	0
miR-425-5p	1256	1249	1076	1122	966	0
miR-451a	0	0	194,289	267,727	46,810	40,211
miR-484	6203	5410	2543	1564	4918	5300
miR-486-5p	0	146	3971	5677	1179	0
miR-532-3p	354	254	1239	2490	0	1400
miR-532-5p	251	258	0	0	0	0
miR-574-3p	5199	3713	48,691	74,742	25,101	48,180
miR-574-5p	20,670	18,789	180,234	217,257	78,303	149,525
miR-615-3p	1621	944	0	0	1150	0
miR-652-3p	1455	1110	0	860	927	0
miR-671-5p	431	149	0	0	0	0
miR-6766-3p	3846	222	0	1818	0	0
miR-7-1-3p	346	307	0	0	0	0
miR-744-5p	254	141	0	0	0	0
miR-7-5p	9101	10,666	3405	897	6088	3271
miR-877-5p	284	0	0	0	0	0
miR-92b-3p	1302	1503	0	866	1478	1527
miR-93-5p	3329	3218	3205	2549	3023	3064
miR-96-5p	1243	965	0	0	0	0
miR-99a-5p	7734	8614	2852	1730	5380	5383
miR-99b-3p	550	288	0	0	0	0
miR-99b-5p	6388	5799	2252	1342	5311	4932

## Data Availability

All data have been made available in the document.

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
