# Peer review of "Lung Fibrosis Is Improved by Extracellular Vesicles from IFNγ-Primed Mesenchymal Stromal Cells in Murine Systemic Sclerosis"

_cells, 2021, doi:10.3390/cells10102727_

Round 1
Reviewer 1 Report
In this paper, authors found increasing the dose of MSC-EVs did not add benefit to the dose previously reported to be efficient in SSc. IFNγ pre-activation improved MSC-EVs-based treatment, essentially in the lungs, with no major impact in the skin. Low dose of IFNγ decreased the expression of fibrotic markers while high dose improved remodeling and anti-inflammatory markers. IFNγ pre-activation upregulated iNos, IL1ra and Il6 in MSCs and ssEVs and, the PGE2 protein in lsEVs. IFNγ-pre-activation improved the therapeutic effect of MSC-EVs preferentially in the lungs of SSc mice by modulating anti-inflammatory and anti-fibrotic markers.
I have some questions.
1) What is the main cause of the difference in effects on the lungs and skin?
2) Why did you focus on IFN-γ?
What results would be expected if MSCs were pre-activated with other inhibitory cytokines, such as IL-10, or pro-inflammatory cytokines, such as IL-6?
3) What kind of the actual situation in vivo, do the difference in results at the mouse level between high and low doses of IFN-γ reflect? For example, high dose corresponds to local site of inflammation, and low dose to areas other than the local area of inflammation?
4) In this paper, several known immunosuppressive markers in IFNγ were pre-activated MSCs. Which molecules are thought to contribute the most?
Author Response
1) What is the main cause of the difference in effects on the lungs and skin?
The most likely explanation is that a systemic administration of EVs will preferentially target the lungs as already reported with MSCs and therefore act more efficiently on the main target organ.
2) Why did you focus on IFN-γ?
What results would be expected if MSCs were pre-activated with other inhibitory cytokines, such as IL-10, or pro-inflammatory cytokines, such as IL-6?
We focused our attention on IFN-γ, which is the main recognized factor as able to prime the immunosuppressive function of MSCs and currently used in many studies. As examples, please find the following reference of a relevant review of literature: de Cássia Noronha et al. Stem Cell Res Ther. 2019; 10: 131
3) What kind of the actual situation in vivo, do the difference in results at the mouse level between high and low doses of IFN-γ reflect? For example, high dose corresponds to local site of inflammation, and low dose to areas other than the local area of inflammation?
The reviewer is right: the difference between low and high doses of IFN-γ may reflect local versus systemic inflammatory reactions. It may also reflect inflammatory flares and different disease states.
4) In this paper, several known immunosuppressive markers in IFNγ were pre-activated most?
We found that the immunosuppressive factors known to be increased upon IFN-γ priming (iNOS, IL1RA, IL6, Cox2, Tsg6) are indeed increased in MSCs, whether with low or high dose of IFN-γ. As expected, HGF and TGFβ1 were not regulated by priming. In fact, it is interesting to underline that the main immunosuppressive factors that are increased upon IFN priming and found both at the mRNA level (the present study) and protein level (other studies) in parental MSCs are not reflected at the mRNA level in isolated EVs. This is the case for instance for Cox2, which is increased in primed MSCs and decreased in primed MSC-derived EVs. This has already been described elsewhere but warrants further investigation to better understand the mechanism of action of MSC-derived EVs, which might slightly differ from parental cells.
Reviewer 2 Report
the authors describe that IFNγ pre-activation of MSCs improved the therapeutic effect of EVs essentially in the lungs of SSc-induced mice with no major impact in the skin by modulating anti-inflammatory and anti-fibrotic markers.
The article is quite exceptional, well done , and each experiments is well supported by histological analysis and markers
I suggest to accept in the present form.
Just explain in the methods the reason to choose only this markers: fibrotic (Col1a1), remodeling 199 (Mmp9) and inflammatory (Il1β, Tnfα) markers
Author Response
Just explain in the methods the reason to choose only this markers: fibrotic (Col1a1), remodeling 199 (Mmp9) and inflammatory (Il1β, Tnfα) markers
Several fibrotic, remodelling, inflammatory and oxidative stress-related markers have been analysed in the different experiments. These genes were selected from our previous experiments as the most representative of the different processes that are dysregulated in SSc. A sentence indicating the genes selected for these processes has been added in the Materials and Methods section (see lines 117-118, page 6).
Reviewer 3 Report
Systemic sclerosis (SSc) is an autoimmune disease with a severe prognosis due to generalized fibrosis and vasculopathy. Immunosuppressive drugs and hematopoietic stem cell transplantation could be efficient to slow disease course but these are also associated with severe side effects.
In the present study, the authors investigated the possibility to enhance the therapeutic effect of ssEVs and lsEVs isolated from murine BM-MSCs (mMSC) in the murine model of SSc. They report that IFNγ pre-activation of MSCs improved the therapeutic effect of EVs essentially in the lungs of SSc-induced mice with no major impact in the skin.
This is a solid study that provide significant evidence on the positive effect of MSCs in the lungs of SSc-induced mice. In the DISCUSSION, the aithors could mention other potential clinical applications of MSCs (see, doi: 10.1007/s00395-021-00881-9 doi: 10.1007/s12975-020-00805-0)
Author Response
This is a solid study that provide significant evidence on the positive effect of MSCs in the lungs of SSc-induced mice. In the DISCUSSION, the aithors could mention other potential clinical applications of MSCs (see, doi: 10.1007/s00395-021-00881-9 doi: 10.1007/s12975-020-00805-0)
Thank you for the positive evaluation of the present study. As recommended, we have added a sentence mentioning other therapeutic applications of primed MSCs in the discussion section (please see lines 349-350, page 24).
Round 2
Reviewer 3 Report
The authors have adequtely addressed my concerns, the manuscript can be published in its present form.